# T2G-Reasoner: Deep Reasoning for Text-to-Gloss Translation

## Abstract

In this work, we present T2G-Reasoner, a framework equipped with a reasoning mechanism to improve text-to-gloss translation (T2G), where gloss is a written record of sign language. The reasoning LLMs have achieved remarkable success in a range of NLP tasks, benefiting from their strong generalization capability stemming from pretraining on massive data. However, incentivizing the reasoning capabilities for the T2G task is challenging due to the absence of gloss information in LLMs' pretraining. Considering shared lexical concepts between two languages, we leverage an advanced LLM to extract word-level alignments as the T2G reasoning process. Instead of directly generating sign language gloss, the proposed method structures the model's output into two distinct components, *i.e.*, the word-level alignments and the final gloss translation. T2G-Reasoner adopts a two-stage training strategy, *i.e.*, SFT-based imitation and RL-based exploration. The T2G-Reasoner model is first fine-tuned on the synthetic reasoning data, which establishes a foundational layer of reasoning capability. As the synthetic reasoning data may be of lower quality, the T2G-Reasoner model further leverages the RL algorithm to autonomously discover optimal word-level alignments. Extensive experiments on two benchmark datasets show that the proposed T2G-Reasoner achieves significant performance improvements. Additionally, our T2G-Reasoner exhibits great potential to address out-of-vocabulary (OOV) challenges in T2G.

## 1 Introduction

Sign languages are the primary languages of culturally Deaf communities and a vital means of communication for many deaf individuals. Translation between sign language and spoken language is an important research topic, which bridges the communication gap between the deaf and the hearing (Yin et al., 2021). To capture the linguistic characteristics of sign language, sign language gloss has been widely used as an intermediate step for generating sign language video from spoken language text (Saunders et al., 2020; 2022). Sign language gloss is the transcription of sign languages sign-by-sign, where each sign has a unique identifier. In this work, we focus on the first step of the cascaded sign language generation pipeline, named text-to-gloss translation (T2G), which aims to translate the spoken language text into the sign language gloss.

Text-to-gloss translation is typically viewed as a low-resource sequence-to-sequence mapping problem. Previous methods (Zhu et al., 2023; Walsh et al., 2022; Egea Gómez et al., 2021; 2022) adopt a one-pass pipeline that directly maps spoken language to gloss sequences, relying on implicit data-driven alignment without word-level supervision, as illustrated in Fig. 1 (a). However, due to the high cost of sign language data annotation, scarce training resources hinder models from learning precise cross-lingual alignments, leading to inaccurate translations (De Coster et al., 2023; Zhou et al., 2021). Instead of directly outputting translation, human translators tend to decompose linguistic structures and resolve ambiguities through a reasoning process. Reasoning as a cognitive process plays a central role in many intellectual activities. With the involvement of explicit reasoning, LLMs have recently made significant advancements in natural language processing and related fields (Guo et al., 2025; Jaech et al., 2024). Witnessing the success of LLM with reasoning, we are motivated to bridge this granularity gap while mitigating data scarcity via a reasoning process.

In this work, we present a reasoning-aware T2G framework, named Text-to-Gloss Translation Reasoner (T2G-Reasoner), to simulate the thinking process in human translation, as illustrated in Fig. 1

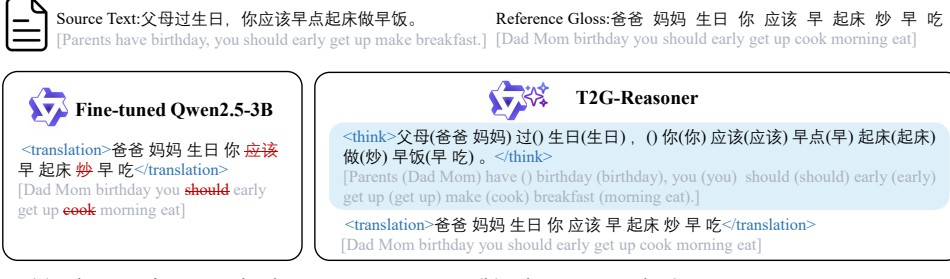

(a) The Previous Method        (b) The Proposed T2G-Reasoner

Figure 1: Comparison between our proposed T2G-Reasoner and previous methods. While previous methods directly generate the translation, our T2G-Reasoner generates both the reasoning process and the final gloss translation. It ensures a more reliable and well-supported output through explicit word-level alignment between two languages. The Chinese example from the CSL-Daily dataset is supplemented with word-by-word English translation in brackets for clarity.

(b). Specifically, the proposed T2G model decomposes the T2G into two stages, *i.e.*, explicit task-aware reasoning and final gloss generation. Our method is inspired by the success of empowering LLMs (Feng et al., 2025; Wang et al., 2024; He et al., 2025) for neural machine translation (NMT) with reasoning capabilities. However, when it goes to T2G, the key challenge becomes how to synthesize the reasoning process beyond the original translation annotation. We notice that, despite grammar gaps, sign language glosses largely share a common vocabulary with spoken language. This inherent lexical commonality enables LLMs to establish semantic-guided alignments at word level, even without gloss-specific expertise.

To empower the T2G model with reasoning ability, the proposed T2G-Reasoner consists of two complementary training strategies, *i.e.*, supervised fine-tuning (SFT-based) imitation and reinforcement learning (RL-based) exploration. Using state-of-the-art commercial models, we enrich the original annotation with synthetic fine-grained correspondences between sign language and spoken language. The T2G-Reasoner model is first fine-tuned on the synthetic thought data using supervised fine-tuning, which serves as a semantic bridge for initial alignment. This enables the T2G-Reasoner model to execute systematic reasoning procedures, ensuring logical coherence and translational accuracy in its outputs. Since incorrect rationales can lead to incorrect final predictions, it is critical to ensure that the rationales produced by LLMs are valid . To address pseudo-reasoning noise, we further adopt reinforcement learning to guide the model to autonomously discover an optimal T2G thought process based on the original gloss annotation.

Combining imitation learning and reinforcement learning strategies, our T2G-Reasoner model achieves better performance over the previous methods on two public benchmarks, *i.e.*, CSL-Daily (Zhou et al., 2021) and PHOENIX14T (Camgoz et al., 2018). Surprisingly, experiments show that translation accuracy on out-of-vocabulary (OOV) glosses is promisingly improved. The OOV glosses denote that the gloss does not appear in the training set but in the evaluation set. We conjecture that this benefits from two factors, *i.e.*, the general linguistic understanding of LLMs and explicit exploration of word correspondences.

Our main contributions are summarized as follows:

- We propose a framework named T2G-Reasoner, which is equipped with a reasoning mechanism. Considering shared lexical concepts between two languages, we leverage the advanced LLM to synthesize word-level alignments as the reasoning process.

- We optimize our T2G-Reasoner model by imitating the synthetic reasoning process, while also encouraging it to explore more accurate word-level alignments. It is feasible to address OOV challenges, which is overlooked in prior research.

- We conduct extensive experiments to validate our approach, which shows encouraging performance improvement on the two public benchmarks, *i.e.*, CSL-Daily (Zhou et al., 2021) and PHOENIX14T (Camgoz et al., 2018).

## 2 RELATED WORK

**Text-to-Gloss Translation.** Camgoz et al. (2018) publish the first sign language neural dataset PHOENIX14T and pioneer the linguistic research for sign language (De Coster et al., 2021; Cao et al., 2022). Most of the previous works typically treat the text-to-gloss translation as a subtask of generation. They (Stoll et al., 2020; Saunders et al., 2020) directly adopt the baseline approaches in NMT. Li et al. (2022) initially focuses on the T2G task, defining it as a low-resource sign language translation task. Considering gloss as a text simplification, they propose a novel editing agent. Instead of directly generating the sign language gloss, the agent predicts and executes the editing program for the input sentence to obtain the output gloss. By leveraging the linguistic feature embedding, Egea Gómez et al. (2021) achieve remarkable performance improvement. Egea Gómez et al. (2022) further apply the transfer learning strategy result in continues performance increasing. Zhu et al. (2023) first introduce effective neural machine translation techniques to T2G with outstanding performance improvements, which lays a good foundation for further research. By iteratively annotating and learning from the synthetic data. Yao et al. (2024) introduce large-scale unlabeled data into T2G training.

**Deep Reasoning.** Reasoning is a fundamental aspect of human intelligence and plays a crucial role in activities such as problem solving, decision making, and critical thinking. The pioneering advancements in reasoning-based LLMs, such as OpenAI's O1 (Jaech et al., 2024) and DeepSeek-R1 (Guo et al., 2025), have excelled in many NLP tasks. Earlier exploration focuses on using inference-time reasoning for solving complex tasks such as math and coding (Qin et al., 2024; Zhang et al., 2024). Recently, there has been a trending belief towards utilizing reasoning-based LLMs for general tasks, such as open-ended text generation (Zhao et al., 2024b), financial tasks (Chu et al., 2025), and machine translation (Wang et al., 2024; Feng et al., 2025). In the machine translation task, Feng et al. (2024) introduce an API-based self-correcting framework. DRT (Wang et al., 2024) utilizes a multi-agent mechanism to distill the structured reasoning process for English-Chinese literature translation. Inspired by DeepSeek-R1-Zero, Feng et al. (2025); He et al. (2025) enhance the translation performance by leveraging the reinforcement learning algorithm. The success of these strategies largely depends on the LLMs' strong generalization capability derived from massive pretraining data.

Different from the aforementioned methods, we focus on incorporating the reasoning process into the text-to-gloss translation. Considering the high lexical similarity between two languages, we leverage an advanced LLM to synthesize the word-level alignments as a T2G reasoning process. To reduce the negative impact of noise in the synthetic reasoning process, we further adopt the RL algorithm to bypass the strict supervision for reasoning processes.

## 3 METHODOLOGY

In this section, we first introduce the overview of our T2G-Reasoner in Sec. 3.1. Then, we elaborate the construction of the T2G reasoning dataset in Sec. 3.2. Finally, we detail the training strategy in Sec. 3.3.

### 3.1 OVERVIEW

The primary objective of the T2G model is to acquire knowledge about the mapping $\mathcal{X} \mapsto \mathcal{Y}$, where $\mathcal{X}$ and $\mathcal{Y}$ denote the collection of spoken language texts and sign language glosses, respectively. Given a set $\mathcal{D} = \{(X^i, \hat{Y}^i)\}_{i=1}^N$ of $N$ labeled samples, a standard T2G model is trained to generated sign language gloss $Y^i$ based on spoken language text $X^i$. The conditional probability of output generation is formulated as:

$$P(Y^i|X^i;\theta) = f(X^i;\theta), \tag{1}$$

where $\theta$ is the parameters of the T2G model.

As shown in Fig. 2, in this work, we aim to improve T2G by simulating the thinking process in human translation. Unlike previous methods, where the output is a gloss translation, we structure the T2G model's output into two distinct components, *i.e.*, the word-level alignment reasoning $R^i$ and the final gloss translation $Y^i$. With the reasoning process, the conditional probability of output

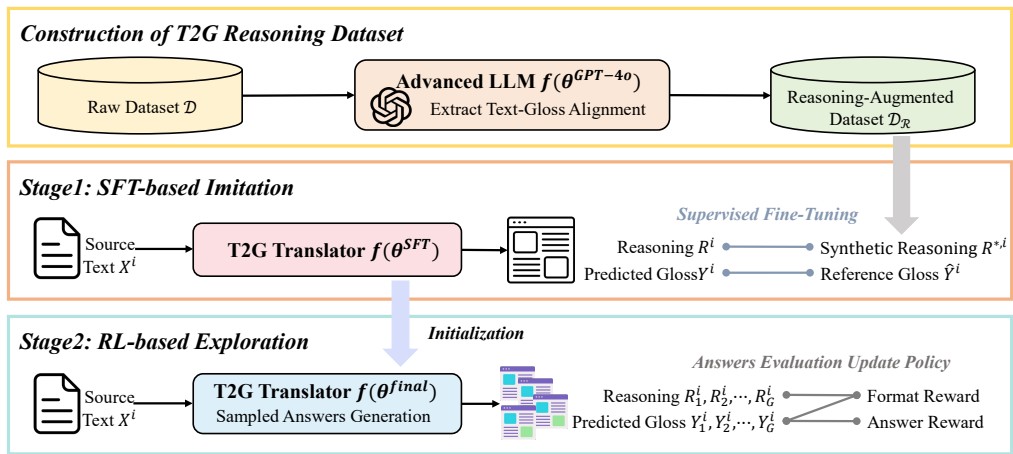

Figure 2: Overview of the proposed T2G-Reasoner. Using the state-of-the-art commercial LLM, *i.e.*, GPT-4o, we first synthesize the reasoning-augmented dataset $\mathcal{D}_{\mathcal{R}}$ by extracting word-level alignment between the two languages as a reasoning process. The T2G-Reasoner model $f(\theta)$ is first fine-tuned on this dataset $\mathcal{D}_{\mathcal{R}}$ supervised by the synthetic reasoning process $R^{*,i}$ and the annotated gloss $\hat{Y}^i$. To mitigate the noise in the synthetic data, the SFT-tuned T2G-Reasoner $f(\theta^{SFT})$ is further optimized by rewarding the correct output. Combining all components, we obtain the proposed T2G-Reasoner $f(\theta^{final})$.

generation (*i.e.*, Equ. 1) is reformulated as:

$$P(R^i, Y^i | X^i; \theta) = f(X^i; \theta). \tag{2}$$

To incentivize the reasoning capabilities of the LLM-based translator, we begin by synthesizing the reasoning process $R^{i,*}$ based on the parallel sample $(X^i, Y^i)$ within the raw dataset $\mathcal{D}$. The raw dataset $\mathcal{D}$ is built upon all training samples. With the calibrated template, we use the advanced LLM, *i.e.*, GPT-4o (gpt, 2023) to extract word-level alignment as the reasoning process. Learning the synthetic reasoning process based on supervised fine-tuning (SFT) enables the T2G-Reasoner with basic reasoning capabilities for T2G. However, without the expert knowledge, the synthetic data inevitably contains mismatches. To mitigate this, the SFT-tuned T2G-Reasoner is further optimized by the reward calculated based on the annotated gloss, leading to free exploration of both reasoning and gloss.

## 3.2 CONSTRUCTION OF T2G REASONING DATASET

Training the T2G-Reasoner model requires a high-quality reasoning dataset. In the pursuit of mimicking the human reasoning process in manual translation, we designed a template that incorporates the reasoning process into the raw T2G dataset $\mathcal{D}$. For each samples $(X^i, Y^i)$ in raw dataset $\mathcal{D}$, the advanced LLM,*i.e.*, GPT-4o $f(\theta^{GPT-4o})$ is required to extract the word-level alignments $R^{*,i} = \{r_1^i, r_2^i, \ldots, r_{T_y}^i\}$ based on the text $X^i = \{x_1^i, x_2^i, \ldots, x_{T_x}^i\}$ and the annotated gloss $\hat{Y}^i = \{\hat{y}_1^i, \hat{y}_2^i, \ldots, \hat{y}_{T_y}^i\}$. To guide the advanced LLM to generate accurate word-level alignments, we manually create an alignment template $T^i = \{x_1^i(), x_2^i(), \ldots, x_{T_y}^i()\{\}\}$, which adds a parentheses bracket after each word of the text and a bracket in the end. The gloss is filled in the corresponding places as the reasoning process $R^{*,i}$. The prompt is displayed in Appendix A.2. In this way, we construct the reasoning-augmented dataset $\mathcal{D}_{\mathcal{R}}$.

## 3.3 TRANSLATION REASONING LEARNING

The T2G-Reasoner model is initialized by the open-source LLM. To incorporate the reasoning capability into the T2G-Reasoner model, our reasoning learning method involves two primary components, *i.e.*, SFT-based imitation and RL-based Exploration.

### 3.3.1 SFT-BASED IMITATION

We provide the T2G-Reasoner model with initial task knowledge in an SFT-based imitation learning strategy, where the model efficiently learns to imitate the synthetic word-gloss alignment and the annotated gloss translation. The two components are concatenated by special tokens as the format reward in the RL algorithm required, which places its reasoning process $R$ within $< \text{think} >$ and $< /\text{think} >$ tags and provide the final translation $Y$ inside $< \text{translation} >$ and $< /\text{translation} >$ tags (The details are in Appendix A.3). Based on the reasoning-augmented dataset $\mathcal{D}_{\mathcal{R}}$, the training objective maximizes the likelihood of the concatenation of the synthetic reasoning process $R^*$ and the annotated gloss $\hat{Y}$, which is formulated as:

$$L(\theta^{SFT}) = -\sum_{i=1}^{N} logP(R^{*,i}, \hat{Y}^i | X^i; \theta^{SFT}). \tag{3}$$

The fine-tuning process establishes a foundational layer of reasoning capability that is critical for the subsequent phase of enhancing the reasoning performance.

### 3.3.2 RL-BASED EXPLORATION

In practice, due to the lack of expert annotation, we note that the synthetic reasoning data inevitably contained noise. To alleviate this issue, we further adopt the RL method to explore more accurate text-gloss alignment based on gloss-specialized knowledge in the SFT-tuned T2G-Reasoner model $f(\theta^{SFT})$.

**Reward Modeling.** To effectively guide the model's reasoning and translation quality, the proposed reward consists of two parts, *i.e.*, the format reward and the answer reward, following the previous work Feng et al. (2025); He et al. (2025) in NMT. These reward are designed to align the model's output with the desired reasoning format and translation quality, respectively. For the format reward, we use regular expression extraction to enforce a structured response format, which is formulated as:

$$S_{format} = \begin{cases} 1 & \text{if format is correct,} \\ -1 & \text{if format is incorrect.} \end{cases} \tag{4}$$

For the answer reward, it indicates the translation quality in the model's output. Specifically, we compute the reward by evaluating the BLEU-4 (Papineni et al., 2002) score of the generated gloss and the corresponding annotated gloss, which is formulated as:

$$S_{answer} = BLEU(Y, \hat{Y}), \tag{5}$$

where $BLEU(\cdot)$ denotes normalized BLEU-4 score. $Y$ and $\hat{Y}$ denote the generated and annotated gloss translation, respectively. It evaluates translation quality by measuring the difference (lexical overlap) between the two sequences, widely used in T2G. The answer reward $S_{answer}$ ranges from 0 to 1 based on the translation quality. The final reward combines both the format reward $S_{format}$ and the metric reward $S_{answer}$, which is formulated as:

$$S = \begin{cases} S_{format} + S_{answer} & S_{format} = 1, \\ -3 & S_{format} = -1. \end{cases} \tag{6}$$

The final reward can vary from 1 to 2 when the output format is correct, and is $-3$ otherwise.

**RL Algorithm.** We use the Group Relative Policy Optimization (GRPO) algorithm to continually optimize the translator based on the generated gloss quality. In each training step, for each text $X^i$ in the training set, we sample a group of $G$ candidate outputs $\{O_1^i, O_2^i, \cdots, O_G^i\}$ from the T2G-Reasoner named as policy model $\pi_{\theta_{old}}$. The corresponding rewards $\{s_1^i, s_2^i, \cdots, s_G^i\}$ are calculated based on the final reward (as in Equ. 6). For each output $O_j^i$ within the group, the advantage $A_j^i$ is computed as $A_j^i = \frac{s_j^i - mean\{s_1^i, s_2^i, \cdots, s_G^i\}}{std\{s_1^i, s_2^i, \cdots, s_G^i\}}$. The goal of the RL objective is to maximize the expected reward, as follows:

$$J_{\text{GRPO}}(\theta) = \mathbb{E}_{X^i \sim P(\mathcal{X}), \{O_j^i\}_{j=1}^G \sim \theta_{\text{old}}(\mathcal{O}|X^i)}$$

$$\left[ \frac{1}{G} \sum_{j=1}^{G} \min\left( \frac{\pi_\theta(O_j^i \mid X^i)}{\pi_{\theta_{old}}(O_j^i \mid X^i)} A_j^i, \ clip\left( \frac{\pi_\theta(O_j^i \mid X^i)}{\pi_{\theta_{old}}(O_j^i \mid X^i)}, 1 - \varepsilon, \ 1 + \varepsilon \right) A_j^i \right) - \beta \, D_{\text{KL}}\left( \pi_\theta \, \| \, \pi_{\theta_{old}} \right) \right],$$

$$\tag{7}$$

where $\varepsilon$ and $\beta$ are hyper-parameters controlling the PPO clipping threshold and the weight of the Kullback–Leibler (KL) divergence penalty Schulman et al. (2017); Shao et al. (2024), respectively.

Combining the two-stage optimization, we obtain the final T2G-Reasoner $f(\theta^{final})$. In this way, we encourage the T2G-Reasoner to not only exploit the reasoning knowledge from the advanced LLM, but also to explore more accurate word-gloss alignments for the correct gloss translation.

## 4 EXPERIMENTS

### 4.1 EXPERIMENTAL SETUP.

**Datasets.** We evaluate our approach on two public sign language translation datasets, *i.e.*, PHOENIX14T (Camgoz et al., 2018) and CSL-Daily (Zhou et al., 2021). Both datasets provide the sign language video, sign language gloss, and spoken language text annotated by human translators. The PHOENIX14T and CSL-Daily datasets collect German and Chinese sign language, respectively. The statistics of the data mentioned above are shown in Appendix A.1.

**Evaluation metrics.** Referring to the previous works (Zhou et al., 2021; Li et al., 2022; Zhu et al., 2023), we evaluate the performance of the generated gloss based on BLEU (Papineni et al., 2002) and ROUGE (Lin, 2004), respectively. For both the evaluation metrics, the higher value demonstrates better translation performance.

**Implementation settings.** We select the Qwen2.5-base series[1] 3B parameter variant as the starting model for T2G-Reasoner training. In the SFT stage, for PHOENIX14T and CSL-Daily dataset, we use the Adam optimizer with a learning rate of $1e-3$ and $2e-3$ for training 3 epochs, respectively. In inference, we use the beam strategy (Wu et al., 2016). For both the PHOENIX14T and CSL-Daily dataset, the search width is 3. In the RL stage, our implementation is based on the Verl[2] framework. For GRPO, the KL penalty coefficient $\beta$ in Equ. 7 is set to 0. The impact of whether leveraging the KL constraint is shown in Appendix A.4. We configure a batch size of 8 and utilize 8 rollouts per input within the GRPO algorithm. The learning rate and temperature are fixed to $1e-6$ and $1.0$, respectively. For both training strategies, the maximum generation length for response is capped at $1,024$ tokens. All models are trained for 3 epochs on 4 NVIDIA 3090 GPUs for about 40 hours of computational time.

### 4.2 COMPARISON WITH STATE-OF-THE-ART METHODS.

We compare the proposed T2G-Reasoner with the previous text-to-gloss approaches on two public benchmarks,*i.e.*, PHOENIX14T (Camgoz et al., 2018) and CSL-Daily (Zhou et al., 2021). The performances are shown in Tab. 1 and Tab. 2, respectively.

As our goal is to explore how to incorporate a reasoning process for T2G, our baseline follows the standard T2G pipeline and adopts the fine-tuned Qwen2.5-3B LLM on the original golden data without the reasoning process. By combining all proposed components, our T2G-Reasoner achieves substantial improvements against the baseline across all evaluation metrics. The T2G-Reasoner achieves 29.05 and 28.40 BLEU-4 on the DEV set of the PHOENIX14T and CSL-Daily dataset, respectively. The quantitative results demonstrate the effectiveness of incentivizing the reasoning process and designs in our T2G-Reasoner.

Recently, Zhu et al. (2023) provide translation performance in different settings, including semi-supervised, transfer learning, and multilingual. Yao et al. (2024) utilize the large-scale monolingual data for T2G. The results prove the advantage of our novel design, which distinguishes our approach from previous T2G methods. As shown in Tab. 2, there are only limited methods that are tested on the Chinese sign language. To attract more research attention to Chinese sign language and other sign languages, we also report our performance on this dataset.

---

[1]https://huggingface.co/Qwen

[2]https://github.com/volcengine/verl

Table 1: Performance comparison of our proposed T2G-Reasoner with methods for T2G on PHOENIX14T.

| | Dev | | | | | Test | | | | |
|---|---|---|---|---|---|---|---|---|---|---|
| | ROUGE | BLEU-1 | BLEU-2 | BLEU-3 | BLEU-4 | ROUGE | BLEU-1 | BLEU-2 | BLEU-3 | BLEU-4 |
| Stoll et al. (2020) | 48.42 | 50.15 | 32.47 | 22.30 | 16.34 | 48.10 | 50.67 | 32.25 | 21.54 | 15.26 |
| Saunders et al. (2020) | 55.41 | 55.65 | 38.21 | 27.36 | 20.23 | 54.55 | 55.18 | 37.10 | 26.24 | 19.10 |
| Amin et al. (2021) | - | - | - | - | - | 42.96 | 43.90 | 26.33 | 16.16 | 10.42 |
| Egea Gómez et al. (2021) | - | - | - | - | - | - | - | - | - | 13.13 |
| Zhang & Duh (2021) | - | - | - | - | - | - | - | - | - | 16.43 |
| Li et al. (2022) | - | - | - | - | - | 49.91 | - | - | 25.51 | 18.89 |
| Saunders et al. (2022) | 57.25 | - | - | - | 21.93 | 56.63 | - | - | - | 20.08 |
| Egea Gómez et al. (2022) | - | - | - | - | - | - | - | - | - | 20.57 |
| Walsh et al. (2022) | 58.82 | 60.04 | 42.85 | 32.18 | 25.09 | 56.55 | 58.74 | 40.86 | 30.24 | 23.19 |
| Zhu et al. (2023) | - | - | - | - | 27.62 | - | - | - | - | 24.89 |
| Yao et al. (2024) | 61.60 | 62.36 | 46.30 | 35.63 | 28.24 | **59.62** | 60.67 | 43.69 | 32.91 | 25.70 |
| Baseline | 61.20 | 62.42 | 45.44 | 34.55 | 27.34 | 58.99 | 60.20 | 42.10 | 30.83 | 23.59 |
| T2G-Reasoner | **62.13** | **64.99** | **47.98** | **36.59** | **29.05** | 59.55 | **63.83** | **45.70** | **34.15** | **26.46** |

Table 2: Performance comparison of our proposed T2G-Reasoner with methods for T2G on CSL-Daily.

| | Dev | | | | | Test | | | | |
|---|---|---|---|---|---|---|---|---|---|---|
| | ROUGE | BLEU-1 | BLEU-2 | BLEU-3 | BLEU-4 | ROUGE | BLEU-1 | BLEU-2 | BLEU-3 | BLEU-4 |
| Li et al. (2022) | - | - | - | - | - | 52.78 | - | - | 29.70 | 21.30 |
| Yao et al. (2024) | 61.52 | 65.88 | 47.67 | 36.05 | 27.95 | 61.75 | 65.88 | 47.90 | 36.06 | 27.74 |
| Baseline | 61.21 | 66.13 | 47.67 | 35.64 | 27.06 | 61.45 | 66.14 | 47.60 | 35.50 | 26.72 |
| T2G-Reasoner | **62.87** | **69.27** | **49.62** | **37.14** | **28.40** | **63.08** | **69.06** | **49.60** | **37.14** | **28.25** |

Table 3: The results of the proposed methods.

| Setting | ROUGE | BLEU-3 | BLEU-4 |
|---|---|---|---|
| Baseline | 61.20 | 34.55 | 27.34 |
| SFT-Tuned | **62.47** | 35.52 | 28.21 |
| T2G-Reasoner | 62.13 | **36.59** | **29.05** |

Table 4: Impact of reward metric selection.

| Accuacy Reward | ROUGE | BLEU-3 | BLEU-4 |
|---|---|---|---|
| BLEU | **62.13** | **36.59** | **29.05** |
| ROUGE | 61.53 | 35.25 | 27.62 |
| BLEU+GOUGE | 61.98 | 35.96 | 28.14 |

### 4.3 ABLATION STUDY

To validate the effectiveness of each component proposed in our T2G-Reasoner framework, unless otherwise specified, we put forward ablation studies on the DEV set of the PHOENIX14T dataset.

**Impact of proposed components.** The main difference between our proposed method and the existing works is that we equip the standard model with a reasoning mechanism. To evaluate the effectiveness of each proposed component, we gradually add them to the baseline T2G model. Directly applying the synthetic reasoning data to the baseline T2G model using SFT delivers a performance gain of $0.84$ BLEU-4. We further apply the group relative policy optimization GRPO algorithm to enforce the predictions under the supervision of the gloss annotation, which further achieves a similar gain. The results are shown in 3.

**Impact of reward metric selection.** As designed in Sec. 3.2, we leverage the BLEU-4 metric as the answer reward, which is always leveraged as the evaluation metric for T2G. In other experiments, the answer reward adopts the BLEU-4 score. The ROUGE is another commonly used translation metric. Since the reward choice significantly affects the learning target and the final outputs, we also experiment with using ROUGE scores and a combination of ROUGE and BLEU-4 as the answer reward in the GRPO algorithm. As shown in Tab. 4, the best results are based on adopting the BLEU-4 metric as the answer reward. As Zhang et al. (2023); Yin et al. (2021) note that the current evaluation for T2G may not be aligned with human judgment, we also believe that how to evaluate the sign language translation quality is an ongoing research topic.

Table 5: Performance comparison of different training paradigms. For supervised fine-tuning, 'w/' and 'w/o' denote whether the base model is trained on the raw data or the reasoning-augmented data. For reinforcement learning, 'w/' and 'w/o' denote whether the format reward includes the reasoning pattern.'S' denotes the setting index.

| S | Supervised Fine-Tuning | | Reinforcement Learning | | ROUGE | BLEU-1 | BLEU-2 | BLEU-3 | BLEU-4 |
| | w/ | w/o | w/ | w/o | | | | | |
|---|---|---|---|---|---|---|---|---|---|
| A | | ✓ | | | 61.20 | 62.42 | 45.44 | 34.55 | 27.34 |
| B | | | ✓ | | 54.63 | 56.97 | 38.84 | 28.04 | 21.17 |
| C | | ✓ | | ✓ | 61.88 | 64.85 | 47.68 | 36.11 | 28.23 |
| D | ✓ | | ✓ | | **62.13** | **64.99** | **47.98** | **36.59** | **29.05** |

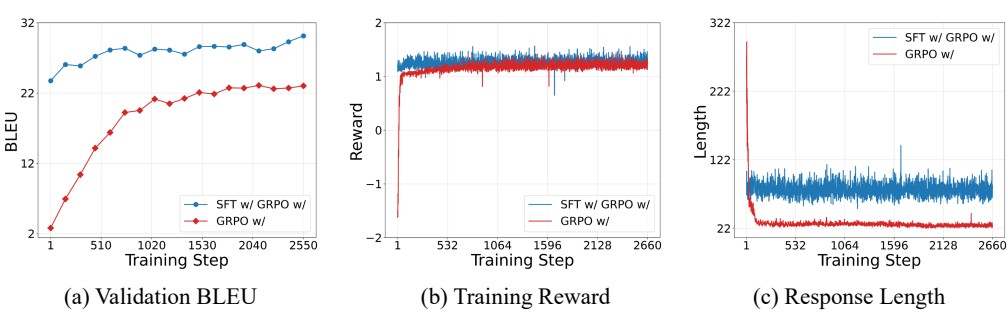

| (a) Validation BLEU | (b) Training Reward | (c) Response Length |
|---|---|---|

Figure 3: Training/validation curves with different training paradigms.

**Performance comparison of different training paradigms.** To determine whether the performance gains stem from the proposed method, we conduct a group of experiments by adopting different training paradigms. The results are shown in Tab. 5. **(i) The effectiveness of fine-tuning on synthetic reasoning data.** To evaluate this, we optimize the base LLM to generate reasoning processes directly based on the RL optimization (Setting B). Since the base LLM lacks gloss-specialist knowledge, it is hard to incentivize the reasoning capability through RL optimization. It only achieves 21.17 BLEU-4, which greatly lags behind the baseline method (A-B). As shown in Fig. 3, we found that the average response length and the training loss of setting B are conveyed within initial training steps. While the model easily masters the required output format, its attempts at reasoning without synthetic data fine-tuning are superficial. It often generates English phrases unrelated to T2G (as examples shown in Appendix A.5). This meaningless reasoning process directly leads to a significantly low translation quality. **(ii) The effectiveness of incorporating the reasoning mechanism for T2G.** To prove final performance gains stemming mainly from the reasoning mechanism instead of the RL optimization, we replace the reasoning process in both SFT and RL (Setting D). It achieves sight performance improvements against the baseline but still lags behind the proposed T2G-Reasoner by 0.82 BLEU-4(A-C-D).

**Impact of LLMs with different parameters.** The effectiveness and training behavior of the proposed T2G-Reasoner are significantly influenced by the base LLM. To evaluate the scalability of the proposed T2G-Reasoner, we conduct three sets of experiments by changing the base LLM with different parameters (0.5B, 1.5B, 3B) from the Qwen2.5 series. As shown in Tab. 6, in all parameter settings, the T2G-Reasoner achieves similar performance gain against the baseline model. This proves the effectiveness of incorporating the reasoning process for T2G. These also prove that using a high-quality base LLM has the potential to achieve further quality gains.

**RL-based exploration analysis.** To provide a more intuitive evaluation of the reasoning self-evolution mechanism, we conduct a comparison between the synthetic and the RL-refined word-level alignments on a training sample. As shown in Fig.4, when a mismatch occurs in the synthetic reasoning process, based on the GRPO optimization, the T2G-model is encouraged to generate the refined reasoning process. The proposed method has the potential to surpass the synthetic data. We highlight the corresponding parts in colors.

**Translation accuracy of low-frequency glosses (including OOV).** As word translation is an essential task for T2G, its accuracy has a significant impact on translation quality. In a data-driven

Table 6: Impact of LLMs with different parameters.

| base LLM | Method | ROUGE | BLEU-1 | BLEU-2 | BLEU-3 | BLEU-4 |
|----------|--------|-------|--------|--------|--------|--------|
| Qwen2.5-0.5B | Baseline | 60.45 | 61.97 | 44.38 | 33.16 | 25.66 |
| | T2G-Reasoner | **60.80** | **63.78** | **45.81** | **34.37** | **26.76** |
| Qwen2.5-1.5B | Baseline | 61.00 | 63.21 | 45.15 | 33.31 | 25.73 |
| | T2G-Reasoner | 61.34 | **64.17** | **47.27** | **35.93** | **28.48** |
| Qwen2.5-3B | Baseline | 61.20 | 62.42 | 45.44 | 34.55 | 27.34 |
| | T2G-Reasoner | 62.13 | **64.99** | **47.98** | **36.59** | **29.05** |

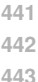
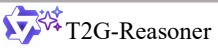

**Expert**

**Source:**
am mittwoch mal sonne mal wolken
hier und da regnet es einzelne
gewitter daran ändert sich auch
am donnerstag nur wenig.
[on Wednesday sometimes sun sometimes
clouds here and there rains it individual
thunderstorms about-this changes itself also
on Thursday only little]
**Reference:**
mittwoch sonne wolke bisschen
regen teil gewitter auch bis
donnerstag gleich weiter
[Wednesday sun cloud some rain part
thunderstorms also until Thursday same
continue]

**GPT-4o**

**Pseudo Reasoning:**
am() mittwoch(mittwoch) mal()
sonne(sonne) mal() wolken(wolke)
hier() und() da() regnet(bisschen)
es(regen) einzelne(teil)
gewitter(gewitter) daran()
ändert() sich() auch(auch) am()
donnerstag(donnerstag) nur()
wenig(gleich)
[at() Wednesday(Wednesday) sometimes()
sun(sun) sometimes() clouds(cloud) here()
and() there() rains(some) it(rain)
individual(part)
thunderstorms(thunderstorms) at-this()
changes() itself() also(also) at()
Thursday(Thursday) only() little(same)]

**T2G-Reasoner**

**Assistant:**
<think>am() am() mittwoch(mittwoch)
mal() sonne(sonne) mal()
wolken(wolke) hier() und() da()
regnet(regen) es() einzelne()
gewitter(gewitter) daran()
ändert() sich() auch() am()
donnerstag(donnerstag) nur()
wenig(wenig) .</think>
[at() Wednesday(Wednesday)
sometimes() sun(sun) sometimes()
cloud(cloud) here() and() there()
rains(rain) it() individual()
thunderstorms(thunderstorms) at-this()
changes() itself() also() at()
Thursday(Thursday) only() little(little) .]

Figure 4: An example illustrates the effect of RL-based exploration to mitigate the noise in the synthetic reasoning process. The German example from the PHOENIX14T dataset is supplemented with word-by-word English translation in brackets for clarity.

Table 7: Translation accuracy of low-frequency glosses. 'Appearance' represents how many times the glosses appear in the TRAIN set of golden parallel data. 'Amount' denotes how many samples contain the low-frequency glosses in the DEV set.

| Appearance | 0 (OOV) | $\leq 1$ | $\leq 3$ | $\leq 5$ | $\leq 7$ | $\leq 9$ |
|-----------|---------|----------|----------|----------|----------|----------|
| Amount | 14 | 10 | 39 | 60 | 72 | 86 |
| Yao et al. (2024) | 0 | 0 | 2.56 | 5.00 | 6.94 | 5.81 |
| Baseline | 0 | 20.00 | 17.95 | 20.00 | 23.61 | 23.26 |
| T2G-Reasoner | 14.29 | 30.00 | 28.21 | 26.67 | 29.17 | 26.74 |

manner, the model tends to predict the glosses with high frequency in the training data. We believe that explicitly predicting the word-level alignments can mitigate this bias. To verify this, we adopt the translation accuracy metric defined in the previous method Yao et al. (2024). The accuracy is formulated as $N_{pred}/N_{all}$, where $N_{pred}$ and $N_{all}$ denote the number of samples that are predicted with the correct gloss and samples that contain the gloss, respectively. Yao et al. (2024) leverages the synthetic T2G data annotated by the fixed rules, which are summarized by human translators, leading to high translation quality on low-frequency glosses. As shown in Tab. 7, our approach achieves high accuracy on low-frequency glosses against the previous method and the baseline method. We also achieve promising performance on the OOV gloss for the first time, based on the general linguistic understanding of LLMs and explicit exploration of word correspondences.

## 5 CONCLUSION AND FUTURE WORK

In this work, we reveal the benefits of performing text-to-gloss translation with explicit reasoning. We present T2G-Reasoner, an effective framework equipped with a reasoning mechanism. Based on the high similarity between the text and gloss, we leverage the advanced LLM to extract the text-gloss alignment as the T2G reasoning process. The base LLM is first fine-tuned on the synthetic reasoning process to establish a foundational layer of reasoning capability. To overcome the negative

impact of the noise in synthetic reasoning processes for T2G, we further leverage the RL algorithm to bypass the strict supervision on synthetic reasoning. The proposed approach achieves promising performance gains in both translation and OOV challenges.

In the future, we are interested in exploring extending the proposed reasoning mechanism to other sign language tasks, such as direct text-to-video generation and video-to-text translation, building upon the strong foundations established in (Zhao et al., 2024a; Chen et al., 2025; Zuo et al., 2025). We also aim to formally study the explicit reasoning traces as a source of model interpretability, providing valuable insights into the decision-making process for more challenging scenarios.

## 6 ETHICAL CONSIDERATION

Despite the growing focus on signed language research, it remains significantly imbalanced compared to spoken languages. As studies in Yin et al. (2021); Bragg et al. (2019); Desai et al. (2024), current efforts are marked by imbalance that risk marginalizing the communities. These include a narrow focus on a limited number of sign languages, overemphasis on translation application at the expense of linguistic inquiry, and a risk of hearing-led research direction.

We contend that building effective and responsible technology necessitates a profound collaborative effort. This must actively involve diverse stakeholders, including Deaf individuals (both native and non-native signers), sign language linguists, and technology developers. In our own work, we are committed to this principle by seeking ongoing feedback from Deaf collaborators and sign language experts throughout the research and development process. We echo the call from the community Yin et al. (2021); Desai et al. (2024) for greater inclusion and encourage the field to adopt participatory design frameworks that ensure our technologies are not just built for the Deaf community, but with them.

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

# A   APPENDIX

## A.1   STATISTICS OF SIGN LANGUAGE DATASETS

As shown in Tab. 8 and Tab. 9, we present the key statistics of the PHOENIX14T and CSL-Daily dataset, respectively. The PHOENIX14T dataset is about weather forecasting. The CSL-Daily dataset is screened by its publishing team and is about daily life (shopping, school, travel, etc.).

Table 8: Statistics of the PHOENIX14T dataset.

|  | Text | | | Gloss | | |
|  | TRAIN | DEV | TEST | TRAIN | DEV | TEST |
|---|---|---|---|---|---|---|
| Sentence | 7,096 | 519 | 642 | 7,096 | 519 | 642 |
| Vocabulary | 2,887 | 951 | 1,001 | 1,085 | 393 | 411 |
| Tot. Words | 99.081 | 6,820 | 7,816 | 55,247 | 3,748 | 4,264 |
| Tot. OOVs | - | 57 | 60 | - | 14 | 19 |

Table 9: Statistics of the CSL-Daily dataset.

|  | Text | | | Gloss | | |
|  | TRAIN | DEV | TEST | TRAIN | DEV | TEST |
|---|---|---|---|---|---|---|
| Sentence | 18,401 | 1,077 | 1,176 | 18,401 | 1,077 | 1,176 |
| unique sentences | 6,598 | 797 | 798 | 6,598 | 797 | 798 |
| Vocabulary | 2,343 | 1,358 | 1,358 | 2,000 | 1,344 | 1,345 |
| Tot. Words/Chars | 291,048 | 17,304 | 19,288 | 133,714 | 8,173 | 9,002 |
| Tot. OOVs | - | 64 | 69 | - | 0 | 0 |

## A.2   PROMPT FOR REASONING CONSTRUCTION

The prompt used for synthesizing the reasoning process in Sec. 3.2 is formulated:

---

**Template for extractation word-level alignments between text and gloss**

We are annotating the translation details from the spoken language text to the sign language gloss. You are a translation expert in sign language gloss. The spoken text `src_text` corresponds to sign language gloss `tgt_gloss`. Due to differences in vocabulary and word order between sign language gloss and spoken language text, your task is to identify the correspondence between the spoken language text and the sign language gloss. For spoken words that do not have a specific sign language gloss, no annotation is required. To ensure the accuracy of the correspondence, for the remaining sign language gloss with unclear correspondence, fill in the sign language gloss within the curly brackets in `template`. Fill in the vocabulary corresponding to the sign language gloss within the parentheses in `template`. The output of the analysis should be in one line, maintain the format of `template`, separate the sign language gloss with a single space, and do not include any special symbols.

---

`src_text` and `tgt_gloss` denote the spoken language text and sign language gloss requiring reasoning annotation.

## A.3   TEMPLATE FOR T2G TRANSLATION

We use a structured prompt template as follow:

> **Template for T2G translation**
>
> A conversation between User and Assistant. The User asks for a translation from `src_text` to `tgt_text`, and the Assistant solves it. The Assistant first thinks about the reasoning process in the mind and then provides the user with the final translation. The reasoning process and final translation are enclosed within $< think > < /think >$ and $< translation > < /translation >$ tags, respectively, i.e., $< think >$ reasoning process here $< /think >< translation >$ final translation here $< /translation >$.
> User: `src_text`

`src_language` and `tgt_language` indicate the source and target languages, and `src_text` denotes the source text requiring translation.

### A.4 IMPACT OF LEVERAGING THE KL CONSTRAINT IN GRPO

Since the weight $\beta$ of the KL divergence penalty is an important hyper-parameter in the GRPO algorithm. We put forward the experiments with different $\beta$. As shown in Fig. 5, our approach achieves better performance when $\beta$ is set to 0.

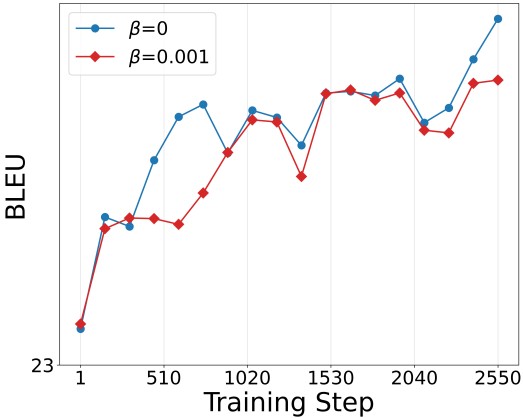

Figure 5: Validation curves with and without the KL divergence.

### A.5 CASE STUDY IN THE RL-ONLY SETTING

As shown in Fig. 6, in the setting of RL-only, the model struggles to align the text and gloss.

**Text:** am tag bestimmen meist wolken das himmelsbild gebietsweise regnet oder schauert es. [at the day determine mostly clouds the sky-image area-wise rains or showers it.]
**Reference:** morgen meistens wolke teil regen schauer [tomorrow mostly cloud part rain shower]
**Thinking:** *maybe about event*
**Hypothesis:** tag wolke himmel regen schauer [day cloud sky rain shower]

**Text:** tiefer luftdruck über nord und mitteleuropa bestimmt auch das wetter bei uns in deutschland. [low air-pressure over North and Central-Europe determines also the weather with us in Germany.]
**Reference:** nord mitte europa enorm tief kommen deutsch land [north central Europe enormous low come Germany]
**Thinking:** *maybe about weather*
**Hypothesis:** tief luft nord mitte europa wetter [low air north central Europe weather]

Figure 6: Case study on the RL-only setting. The German examples from the PHOENIX14T dataset are supplemented with word-by-word English translation in brackets for clarity.

Table 10: The results of the proposed methods on G2T.

| Setting | Dev | | | | | Test | | | | |
|---|---|---|---|---|---|---|---|---|---|---|
| | ROUGE | BLEU-1 | BLEU-2 | BLEU-3 | BLEU-4 | ROUGE | BLEU-1 | BLEU-2 | BLEU-3 | BLEU-4 |
| Baseline | 51.71 | 51.01 | 38.28 | 30.56 | 25.40 | 50.54 | 49.64 | 37.23 | 29.54 | 24.45 |
| SFT-Tuned | 53.19 | 52.35 | 39.59 | 31.65 | 26.31 | 51.53 | 50.61 | 37.90 | 29.92 | 24.67 |
| G2T-Reasoner | **54.30** | **52.91** | **40.11** | **32.08** | **26.70** | **53.46** | **51.98** | **39.53** | **31.60** | **26.29** |

Table 11: The $M$-shot T2G performance based on GPT-4o.

| $M$-shot | ROUGE | BLEU-1 | BLEU-2 | BLEU-3 | BLEU-4 |
|---|---|---|---|---|---|
| 0 (Zero-shot) | 33.23 | 27.37 | 14.63 | 8.28 | 5.12 |
| 1 | 39.01 | 37.50 | 20.72 | 12.30 | 7.77 |
| 3 | **43.89** | 42.62 | 25.10 | 15.93 | 10.52 |
| 10 | 42.07 | **42.64** | **26.13** | **16.88** | **11.24** |

## A.6 THE RELATIONSHIP BETWEEN WORD ALIGNMENT QUALITY AND TRANSLATION ACCURACY

As our approach introduces the word-level alignment as a reasoning process, we provide an evaluation how word alignment quality impacts translation accuracy. Given the lack of word-level alignment annotations in the datasets, we define the alignment quality as word alignment accuracy based on the annotated gloss. This metric measures the proportion of glosses in the model's generated alignment that appear in the GT. The Pearson correlation coefficient is computed between the word alignment accuracy and translation quality (BLEU-4 score) for all samples in the DEV set. The correlation coefficient r is 0.54, which indicates a positive correlation between word alignment accuracy and translation quality.

## A.7 PERFORMANCE ON GLOSS-TO-TEXT TRANSLATION

To verify whether our approach is relevant to the direction of translation, we evaluate the performance on the converse task of T2G on PHOENIX14T. As shown in Tab. 10, we observe that our proposed approach achieves similar performance gains on gloss-to-text translation (G2T) as text-to-gloss translation (T2G). The experimental results further demonstrate the effectiveness of our approach.

## A.8 POTENTIAL RISK OF GENERATING NON-EXISTING GLOSSES

We calculate the proportion of generated glosses that are non-existing on the DEV set of PHOENIX14T. The non-existing glosses generation rate is nearly identical between the baseline model (0.38%, 13 out of 3456) and our T2G-Reasoner (0.37%, 13 out of 3478).

## A.9 GPT-4O FOR T2G

We evaluated the T2G capability of advanced LLMs (i.e., GPT-4o) under both zero-shot and few-shot settings. For few-shot, we provide the GPT-4o with $M$ most similar training samples retrieved through BM25. The results are shown in Tab. 11. Simply increasing the number of reference examples (from 3 to 10) resulted in only a marginal performance gain. Neither setting yielded a satisfactory translation quality. They indicate that LLMs lack a fundamental understanding of cross-lingual alignments for T2G.

The prompt used for Zero-shot T2G is formulated:

> **Propmt for Zero-shot T2G**
>
> The goal is to translate spoken sentences into accurate sign language transcriptions that align with the expression habits of deaf individuals. In these sign language transcriptions, the sign language words are a subset of the spoken words, referencing the semantics of the spoken language, but not entirely consistent with the meanings of the spoken words. Generate an accurate sign language transcription corresponding to the spoken sentence `src_text`, adhering to the expression habits of deaf individuals.

`src_text` denotes the spoken language text.

The prompt used for Few-shot T2G is formulated:

> **Propmt for Few-shot T2G**
>
> Reference sample similar to the spoken sentence `src_text`: `BM_retrieval_info` The goal is to translate spoken sentences into accurate sign language transcriptions that align with the expression habits of deaf individuals. In these sign language transcriptions, the sign language words are a subset of the spoken words, referencing the semantics of the spoken language, but not entirely consistent with the meanings of the spoken words. Additionally, there isn't a complete one-to-one correspondence between words; there may be one-to-many or many-to-one mappings. Specific cases can be referenced in how spoken words are mapped to sign language words in similar samples. There are significant differences in word order between the sign language transcription and the spoken sentence. Specific cases can be compared and referenced in how grammatical components of spoken sentences are converted in terms of word order in similar samples. Before translating, please carefully compare the various parts of the spoken sentence `src_text` with the similar parts in the reference samples, and generate the corresponding accurate sign language transcription that aligns with the expression habits of deaf individuals.

`src_text` and `BM_retrieval_info` denote the spoken language text and its $M$ most similar training samples via the BM25, respectively.

