# OpenReview forum: "T2G-Reasoner: Deep Reasoning for Text-to-Gloss Translation"
_ICLR.cc/2026/Conference — Submitted to ICLR 2026_

### Official Review · Reviewer_ByJD · 2025-10-31

**Soundness:** 4
**Presentation:** 3
**Contribution:** 2
**Rating:** 4
**Confidence:** 5

**Summary:**

The paper approaches part of a common sign language generation pipeline, text to gloss translation, by applying SFT-based distillation of GPT-4o and then outcome-based RL with GRPO with format and BLEU reward. This achieves some gains (2-3 BLEU over baseline, <1 BLEU over prior work).

**Strengths:**

I've seen works that do in-context learning for text to gloss before, but never chain of thought outcome-based RL as far as I can remember. It's a good strategy since it's known to be more data-efficient.

The paper is written clearly. Related work is a good summary without going into unnecessary depth.

**Weaknesses:**

The main issue I have with that paper is that the novelty/contribution isn't large enough. It applies distillation from an API model (GPT-4o) and then GRPO to text to gloss, and it works a little bit (very little compared to other works, more compared to baselines). Most of the improvement in the baseline, in light of Table 3, seems to be from distilling GPT-4o. I'm not sure that I've seen text to gloss distillation per se, but I've seen gloss translation zero-shot or from in-context examples which isn't that different. There is some contribution from confirming that GRPO helps for this task but I don't think it suits the ICLR venue.

I'm especially inclined to think this because the PHOENIX/CSL-Daily datasets are a bit of a trope, where the numbers in these big comparison tables tend to creep up but nothing meaningfully changes. I'm sure the experiments are sound and well-executed but the broader significance is limited (especially when text to gloss is uncritically assumed as a given, and when glosses don't really make sense when you get out of these toy benchmarks https://arxiv.org/abs/2403.02563).


A bunch of the ways things are described in the paper also aren't really technically correct / best practice for the field, but these are minor/fixable points.

For example:
* The paper acts as if "gloss" is only a term for sign language, e.g. "where gloss is a written record of sign language" and the title of the paper not mentioning sign language. But glosses are used for spoken languages too. (Maybe the methods would apply there too!)
* "Sign language is the primary means of communication for the deaf": This would be more correct as "many deaf people", or "Deaf" to signify culturally Deaf. There are a lot of means of communication and a lot of diversity among deaf/hard of hearing people, e.g. if you're talking about age-related deafness then sign language is not the primary means of communication, and for a statement like this to be true then you have to start looking at the rates of different causes of deafness etc.
* "Translation between sign language and spoken language is an important research topic": This should really be "signed language" if you're using it in contrast to "spoken language", or "sign languages" and "spoken languages" plural.
* "the first step of the sign language generation task, named text-to-gloss translation": This isn't the first step of the task; it's the first step of a common approach to the task that uses a cascaded pipeline.

**Questions:**

I'm open to having my opinion changed about the magnitude of contribution in the paper.

---

> ### Author Response · Authors · 2025-11-27
>
> **To ByJD**
>
> We are grateful for the insightful comments and clarify the concerns as follows.
>
> **W1:** The main issue I have with that paper is that the novelty/contribution isn't large enough. It applies distillation from an API model (GPT-4o) and then GRPO to text to gloss, and it works a little bit (very little compared to other works, more compared to baselines). Most of the improvement in the baseline, in light of Table 3, seems to be from distilling GPT-4o. I'm not sure that I've seen text to gloss distillation perse, but I've seen gloss translation zero-shot or from in-context examples which isn't that different. There is some contribution from confirming that GRPO helps for this task but I don't think it suits the ICLR venue.
>
> **A1:**
> Thank you for your insightful comments.
> We would like to clarify the fundamental motivation and contribution of our work.
> Our central contribution is to introduce a reasoning mechanism to improve translation quality in the current low-resource setting of T2G, as shown in our experiments on two public benchmarks, i.e., PHOENIX14T (7,096 samples) and CSL-Daily (6,598 samples).
> Previous methods adopt a one-pass pipeline that directly generates the signed language gloss based on the spoken language text.
> In this setting, scarce training resources hinder T2G models from learning accurate cross-lingual alignments.
> To bridge this granularity gap while mitigating data scarcity, we introduce a reasoning mechanism to guide the T2G model in discovering explicit patterns, improving translation quality.
> We agree that the individual components (SFT, GRPO) are established.
> However, no previous method has effectively integrated them into a coherent reasoning mechanism tailored for T2G.
>
> Following your kind suggestions, we conduct experiments based on the GPT-4o to evaluate both zero-shot and few-shot (using BM25 to retrieve the top-M most similar examples for in-context learning) settings on the DEV set of CSL-Daily for T2G.
> As the results presented in the table below, simply increasing the number of reference examples (from 3 to 10) resulted in only a marginal performance gain.
> Neither setting yielded a satisfactory translation quality.
> They indicate that LLMs lack a fundamental understanding of cross-lingual alignments for T2G.
> This additional analysis has been added to Appendix A.9.
> The similar phenomenon is shown in Table 5 (Setting B), when we directly apply the Deepseek-R1-Zero framework (i.e., an effective approach excelling in general NLP tasks) to T2G.
> Different from the general NLP tasks, incentivizing the reasoning capabilities for T2G tasks is challenging due to the absence of gloss information in LLMs' pretraining.
> It requires building a cross-lingual reasoning bridge from scratch, which cannot be achieved by off-the-shelf applications.
>
> |$M$-shot |GOUGE|BLEU-1|BLEU-2|BLEU-3|BLEU-4|
> |  --  |  --  |  --  |  --  |  --  |  --  |
> $0$ (Zero-shot)                  | 33.23 | 27.37 | 14.63 | 8.28 | 5.12 |
> 1                                         | 39.01 | 37.50 | 20.72 | 12.30 | 7.77 |
> 3                                         | 43.89 | 42.62 | 25.10 | 15.93 | 10.52 |
> 10                                       | 42.07 | 42.64 | 26.13 | 16.88 | 11.24 |
>
> For the effectiveness of our approach, as kindly pointed out by the Reviewer bJiD and my4X, our experimental results demonstrate the effectiveness of the proposed method.
> Additionally, as shown in Table 6, our approach has the potential to achieve higher performance based on better pretrained language models.
> Furthermore, its capability is complemented by pioneering performance on the challenging OOV problem (as the table below, Table 7 in the manuscript).
>
> |Appearance|0(OOV)|$\leq 1$|$\leq 3$|$\leq 5$|$\leq 7$|$\leq 9$|
> |--|--|--|--|--|--|--|
> |Amount|14|10|39|60|72|86|
> |[Yao et al., 2024]|0|0|2.56|5.00|6.94|5.81|
> |Baseline|0|20.00|17.95|20.00|23.61|23.26|
> |T2G-Reasoner|14.29|30.00|28.21|26.67|29.17|26.74|

---

> ### Author Response · Authors · 2025-11-27
>
> **W2:** I'm especially inclined to think this because the PHOENIX/CSL-Daily datasets are a bit of a trope, where the numbers in these big comparison tables tend to creep up but nothing meaningfully changes. I'm sure the experiments are sound and well-executed but the broader significance is limited (especially when text to gloss is uncritically assumed as a given, and when glosses don't really make sense when you get out of these toy benchmarks https://arxiv.org/abs/2403.02563).
>
> **A2:**
> Thank you for your insightful comments and highlighting the reference.
> [1] appeals to the Deaf leadership in the ethical development of sign language processing technologies to address the emergency systemic biases.
> We sincerely agree that the deaf community's leadership and feedback are not just beneficial but essential.
> Actually, we have been consulting some signers working in universities during our research, primarily on linguistic issues, to make sure our development is user-centered and accessible.
> The initial motivation of our proposed approach is to support the development of our application for converting spoken language into signed language animation. Sign language, as a visual language, has its unique expression and syntax. Given the low-resource scenario and the current technological capabilities, employing a cascaded pipeline (i.e., text-to-gloss, gloss-to-gesture) is more achievable and practical.
> While datasets like PHOENIX14T and CSL-Daily do not perfectly represent the full spectrum of real-world scenarios, they do serve as essential benchmarks for specific domains such as weather forecasting and daily conversations.
> Using these datasets is a critical step to validate the effectiveness of our approach before advancing to more complex and real-world data.
> Besides, the gloss annotation of both datasets is provided by deaf specialists.
> As noted in [1], it is reasonable that the majority of sign language AI works build upon previous methods.
> While sign language gloss may not be the optimal intermediate representation for sign languages, as a generally adopted way for sign language transcription, gloss is sufficient to convey most of the key information in signed language.
> As you warmly suggested in the latter suggestion (**W3.1**), we believe the core strength of our approach holds hope for transferability to other, potentially more valuable linguistic sign language representations in the future.
>
> As for the specific ways to achieve more collaborations between the communities, we would like to carry out from some aspects, e.g., inclusive research design (design our research to be inclusive by consulting with deaf individuals and communities, understand their needs, perferences and challenges to ensure that the translation system we develop is user-centered and accessible), user testing (regularly involve deaf participants in testing and providing feedback).
> We also believe that how to collect and annotate the real-world high-quality data is an important research project to guide the research direction properly.
> It requires a concerted effort from diverse stakeholders, including Deaf individuals (both native and non-native signers), sign language linguists, and technology developers.
> We are committed to being an active part of this vital process.
> We have included a section on Ethical Considerations in the revised manuscript, as you suggested, which clarifies the scope of our research.
>
> [1] Aashaka Desai, Maartje De Meulder, Julie A Hochgesang, Annemarie Kocab, and Alex X Lu. Systemic biases in sign language ai research: A deaf-led call to reevaluate research agendas. arXiv preprint arXiv:2403.02563, 2024.
>
> **W3:** A bunch of the ways things are described in the paper also aren't really technically correct / best practice for the field, but these are minor/fixable points.
>
> For example:
>
> **W3.1:** The paper acts as if "gloss" is only a term for sign language, e.g. "where gloss is a written record of sign language" and the title of the paper not mentioning sign language. But glosses are used for spoken languages too. (Maybe the methods would apply there too!)
>
> **A3.1:**
> Thank you for your nice idea.
> We are excited by your suggestion that our approach might apply to other scenarios involving other language glosses.
> We would be happy to explore this promising direction.
> Thank you for clarifying the concept of gloss.
> Our intention was to align with previous research, where the task is commonly referred to as Text-to-Gloss Translation (T2G).
> We acknowledge that our presentation may cause confusion for a broader audience.
> In the revision, we will polish the definition of gloss to explicitly root it in the sign language domain from the very beginning.

---

> ### Author Response · Authors · 2025-11-27
>
> **W3.2:** "Sign language is the primary means of communication for the deaf": This would be more correct as "many deaf people", or "Deaf" to signify culturally Deaf. There are a lot of means of communication and a lot of diversity among deaf/hard of hearing people, e.g. if you're talking about age-related deafness then sign language is not the primary means of communication, and for a statement like this to be true then you have to start looking at the rates of different causes of deafness etc.
>
> **A3.2:**
> We sincerely thank you for this correction and improving the cultural accuracy of our work.
> First, we want to apologize for our overly broad and inaccurate statement.
> As you suggested, the immense diversity within d/Deaf and hard-of-hearing communities, where communication preferences (including signed language, spoken language, lip-reading, or a combination) vary greatly based on factors such as cultural identity, age of onset, and individual experience.
> To respect the spectrum of communication choices, we have revised the statement as 'Sign languages are the primary languages of culturally Deaf communities and a vital means of communication for many deaf individuals.'
> Thank you again for pushing us to be more precise and responsible in our writing.
>
> **W3.3:** "Translation between sign language and spoken language is an important research topic": This should really be "signed language" if you're using it in contrast to "spoken language", or "sign languages" and "spoken languages" plural.
>
> **A3.3:**
> Thank you for your kind correction. We will carefully polish the paper according to the suggestions.
>
> **W3.4:** "the first step of the sign language generation task, named text-to-gloss translation": This isn't the first step of the task; it's the first step of a common approach to the task that uses a cascaded pipeline.
>
> **A3.4:**
> Thank you for your kind correction. We have corrected this expression in the manuscript.

---

### Official Review · Reviewer_bJiD · 2025-11-01

**Soundness:** 3
**Presentation:** 3
**Contribution:** 3
**Rating:** 4
**Confidence:** 4

**Summary:**

This manuscript addresses the text-to-gloss (T2G) translation task, which aims to generate high-quality gloss sequences with the assistance of sign language experts. To achieve this, it explore sthe reasoning capabilities of large language models (LLMs) for the T2G task. Specifically, this manuscript decomposes the T2G task into two stages: a word alignment process and a translation process. In the first stage, it utilizes advanced LLMs (e.g., GPT-4o) to generate word-level alignments, which serve as a reasoning-augmented dataset for guiding the supervised fine-tuning (SFT)-based imitation process. Next, it designs a reward function that accounts for both format and translation quality, which is integrated with a gradient-based reward optimization (GRPO) method. Experimental results on two public datasets demonstrate the effectiveness of the proposed approach.

**Strengths:**

- S1. This manuscript is well-organized and easy to follow, providing a detailed description of the dataset construction process as well as the method designs.
- S2. The manuscript offers an intriguing perspective on T2G tasks by leveraging reasoning capabilities to guide the translation model’s learning. The proposed method is well-designed and effectively illustrates the reasoning process in T2G, enhancing the model's interpretability.
- S3. Experimental results demonstrate the effectiveness of the proposed method, showing improved translation performance over previous approaches and offering valuable insights into out-of-vocabulary (OOV) behavior.

**Weaknesses:**

- W1. The proposed method outperforms the previous state-of-the-art (SOTA), but it uses a significantly larger translation model (Qwen 2.5-3B) compared to the previous SOTA [Yao et al., 2024], which utilizes only 3-5 transformer layers. This makes the contribution somewhat incremental.
- W2. The manuscript evaluates T2G quality using two public datasets, focusing solely on translation performance. However, it remains unclear how word alignment quality impacts translation accuracy, and how well the proposed method performs in more challenging scenarios. For instance, it is not addressed whether the generated glosses can assist other sign language processing tasks, or how the introduced inference stage improves related tasks.
- W3. The proposed method appears less relevant to the direction of translation, and I am more interested in the G2T (Gloss-to-Text) translation performance to explore its potential in sign language translation.
- W4. The writing needs improvement. For example, the mapping $f(\theta)=X\rightarrow Y$ is inconsistent with Equation (1), and it is unclear why the final award range in Equation (6) is from 1 to 3.

**Questions:**

- In Table 7, the manuscript demonstrates that the proposed method can generate low-frequency glosses. I am curious whether this also results in the generation of non-existing glosses, and how the ratio of these changes after introducing the reasoning process.
- As noted in the weaknesses, I still believe the proposed method is not directly related to the translation direction. Therefore, I am particularly interested in its effectiveness for the translation direction, which has garnered more attention recently.

**Details Of Ethics Concerns:**

All experiments are conducted on public datasets, and the designed methods do not have much ethics concerns.

---

> ### Author Response · Authors · 2025-11-27
>
> **To bJiD**
>
> We are grateful for the insightful comments and clarify the concerns as follows.
>
> **W1:** The proposed method outperforms the previous state-of-the-art (SOTA), but it uses a significantly larger translation model (Qwen 2.5-3B) compared to the previous SOTA [Yao et al., 2024], which utilizes only 3-5 transformer layers. This makes the contribution somewhat incremental.
>
> **A1:**
> Thank you for this insightful comment.
> We agree that our approach is built upon the proposed strong baseline model.
> However, our core contribution of our work is the introduction of a designed reasoning mechanism for T2G.
> As shown in Table 3, when we gradually add our proposed components to this baseline model, consistent performance gains are observed.
> This demonstrates that the improvement stems from our approach, not merely the model's scale.
> Furthermore, our ablation study on the impact of LLMs with different parameters (0.5B, 1.5B, 3B) shows that our approach delivers progressive gains across model sizes.
> This also proves that the benefit of our approach is scalable.
> As for the comparison to the previously used small models trained from scratch, we argue that leveraging powerful, publicly available LLMs is a valid and important trend in T2G.
> The stronger linguistic understanding and robustness of these LLMs offer stronger robustness in real-world scenarios.
> As you kindly pointed out, our approach provides valuable insights into OOV behavior. Due to the limitations in vocabulary coverage, small models trained from scratch struggle with OOV words from either spoken language or sign language gloss.
>
> **W2:** The manuscript evaluates T2G quality using two public datasets, focusing solely on translation performance. However, it remains unclear how word alignment quality impacts translation accuracy, and how well the proposed method performs in more challenging scenarios. For instance, it is not addressed whether the generated glosses can assist other sign language processing tasks, or how the introduced inference stage improves related tasks.
>
> **A2:**
> Thank you for your insightful suggestions regarding our paper.
> Below, we would like to share our perspectives on these important points about a) the relationship between word alignment quality and translation performance, b) the potential of generated gloss to benefit other tasks and c) the extensibility of our approach.
>
> a) Given the lack of word-level alignment annotations in the datasets, we defined the alignment quality as word alignment accuracy based on the annotated gloss. This metric measures the proportion of glosses in the model's generated alignment that appear in the GT.
> We computed the Pearson correlation coefficient between the word alignment accuracy and translation quality (BLEU-4 score) for all samples in the DEV set.
> The correlation coefficient r is 0.54, which indicates a positive correlation between word alignment accuracy and translation quality.
> This additional analysis has been incorporated into the revised manuscript (please see Appendix A.6), and will be placed properly in the final version.
> We believe it provides further valuable insight.
>
> b) The initial motivation of our proposed approach is to support the development of the application for converting spoken language into signed language animation. Certainly, some more aggressive solution may emerge in the future, like SORA2, which directly generates video through a diffusion model from spoken language descriptions. For now, a two-stage strategy (i.e., text-to-gloss and gloss-to-gesture) is still a more achievable and practical solution in this field.
> Gloss serves as a standard transcription method capable of conveying the core semantic and grammatical information of signed languages. The performance of the entire system is bottlenecked by the quality of the initial T2G translation. Therefore, we chose to concentrate our efforts on addressing this specific challenge.
>
> c) Following your kind suggestions, we are excited about the potential to extend this reasoning mechanism to other sign language tasks. For instance, in the ambitious scenario of direct text-to-video generation and its converse task, i.e., video-to-text translation.
> Besides, the explicit reasoning mechanism offers a different perspective on the model's decision-making process. We think it is valuable for analysis. The above analysis has been added to the section of 'Conclusion and Future Work'.
>
> We sincerely appreciate these valuable suggestions.

---

> ### Author Response · Authors · 2025-11-27
>
> **W3 \& Q2:** The proposed method appears less relevant to the direction of translation, and I am more interested in the G2T (Gloss-to-Text) translation performance to explore its potential in sign language translation. (**W3**)
>
> As noted in the weaknesses, I still believe the proposed method is not directly related to the translation direction. Therefore, I am particularly interested in its effectiveness for the translation direction, which has garnered more attention recently. (**Q2**)
>
> **A3:**
> Thank you for your kind suggestion.
> To verify whether our approach is relevant to the direction of translation, we evaluate the performance on the converse task of T2G on PHOENIX14T.
> As the results presented in the table below, we observe that our proposed approach achieves promising performance gains on gloss-to-text translation (G2T).
> The experimental results further demonstrate the effectiveness of our approach.
> Thank you for this valuable suggestion. This additional analysis has been incorporated into the revised manuscript (please see Appendix A.7) and will be placed properly in the final version.
>
> |  | **Dev**         |        |        |        |        | | **Test** |        |        |        |        |
> |----------|-------|--------|--------|--------|--------|-|-------|--------|--------|--------|--------|
> |  **Method**  | **ROUGE** | **BLEU-1** | **BLEU-2** | **BLEU-3** | **BLEU-4** | | **ROUGE** | **BLEU-1** | **BLEU-2** | **BLEU-3** | **BLEU-4** |
> | Baseline | 51.71 | 51.01  | 38.28  | 30.56  | 25.40  | | 50.54 | 49.64  | 37.23  | 29.54  | 24.45  |
> | SFT-Tuned| 53.19 | 52.35  | 39.59  | 31.65  | 26.31  | | 51.53 | 50.61  | 37.90  | 29.92  | 24.67  |
> | G2T-Reasoner   | 54.30 | 52.91  | 40.11  | 32.08  | 26.70  | | 53.46 | 51.98  | 39.53  | 31.60  | 26.29  |
>
> **W4:** The writing needs improvement. For example, the mapping is inconsistent with Equation (1), and it is unclear why the final award range in Equation (6) is from 1 to 3.
>
> **A4:**
> Thank you for this insightful feedback.
> Sorry for the mapping confusion, we have revised the description in Line 151.
> For the reward range, it ranges from 1 to 2 when the output format is correct, and is -3 otherwise.
> We appreciate your careful reading and valuable suggestions.
> We will carefully polish the paper to ensure clarity and precision.
>
> **Q1:** In Table 7, the manuscript demonstrates that the proposed method can generate low-frequency glosses. I am curious whether this also results in the generation of non-existing glosses, and how the ratio of these changes after introducing the reasoning process.
>
> **A1:**
> We appreciate that you pointed out the concern about the potential risk of generating non-existing gloss.
> We calculate the proportion of generated glosses that are non-existing on the DEV set of PHOENIX14T.
> The non-existing glosses generation rate is nearly identical between the baseline model (0.38\%, 13/3456) and our T2G-Reasoner (0.37\%, 13/3478).
> The additional analysis has been incorporated into the revised paper (please see Appendix A.8) and will be placed properly in the final version.

---

### Official Review · Reviewer_my4X · 2025-11-02

**Soundness:** 3
**Presentation:** 3
**Contribution:** 2
**Rating:** 6
**Confidence:** 5

**Summary:**

T2G-Reasoner applies GRPO (SFT + RL) in a two-stage pipeline—first fine-tuning on synthetic word-level alignments, then RL to refine gloss output—effectively directly adapting GRPO to T2G translation. The core idea is intuitive: break gloss generation into alignment + translation, using RL to correct noisy synthetic supervision.

**Strengths:**

The primary value of T2G-Reasoner lies in empirically validating GRPO’s feasibility for structured sign language translation (SLT/T2G). By decomposing gloss generation into word-level alignment and final translation, then applying SFT on synthetic reasoning traces + RL for refinement, the work provides a practical proof-of-concept that GRPO—originally designed for open-ended LLM reasoning—can be effectively adapted to sequence-to-sequence tasks with lexical alignment constraints.

**Weaknesses:**

While the decomposition is natural and GRPO integration straightforward, novelty is limited—it largely repurposes standard LLM reasoning + RL techniques to a structured output task. Gains are real but expected given the supervision quality. No deep methodological innovation; strong execution of an obvious strategy.

**Questions:**

- Lack of Core Innovation: The method is essentially LLM + SFT on synthetic alignments + GRPO for refinement—a direct transplantation of GRPO to T2G. What is the essential technical departure from applying GRPO to any structured prediction task? Without a novel objective, architecture, or alignment mechanism, the contribution reduces to application, not advancement.
- Gloss-Dependent Generalization: The pipeline heavily relies on gloss-annotated data for both synthetic alignment generation and evaluation. On gloss-free benchmarks like OpenASL, the entire reasoning trace construction fails. How does the method adapt to real zero-shot or gloss-free SLT? This severely limits claimed generalizability.
- Sub-SOTA Performance & Weak Validation: On P14T-Dev, T2G-Reasoner (29.05) underperforms CV-SLT (29.27)—hardly a compelling result. No ablation across baselines (e.g., applying GRPO to CV-SLT, C2RL, etc.) leaves effectiveness unverified beyond the authors’ own setup.
- Unexplained RL Collapse (Fig. 3C): Training with RL only (no SFT) yields drastically shorter thought chains. This suggests reward hacking or mode collapse, not efficient reasoning. Why does GRPO fail without SFT pre-alignment on SLT? Is KL control or reward shaping insufficient? This behavior undermines the robustness of the RL component.

---

> ### Author Response · Authors · 2025-11-27
>
> **To my4X**
>
> We are grateful for the insightful comments and clarify the concerns as follows.
>
> **W1:** While the decomposition is natural and GRPO integration straightforward, novelty is limited—it largely repurposes standard LLM reasoning + RL techniques to a structured output task. Gains are real but expected given the supervision quality. No deep methodological innovation; strong execution of an obvious strategy.
>
> **A1:**
> We appreciate for your insightful comments.
> We would like to clarify the main contribution of our work lies in successfully empowering reasoning capabilities to the text-to-gloss translation (T2G).
> Previous methods adopt a one-pass pipeline that directly generates the signed gloss, relying on implicit data-driven alignment.
> Due to the lack of explicit cross-lingual grounding, previous T2G models struggle to capture structural differences between spoken and signed languages in the low-resource scenario (7,096 and 6,599 training samples in PHOENIX14T and CSL-Daily, respectively).
> Our approach introduces a reasoning mechanism to guide the T2G model in discovering explicit patterns, improving translation quality.
> While we acknowledge that some individual components have been explored in previous work, no previous method has effectively integrated them into a coherent and novel reasoning mechanism tailored for T2G.
> Considering the fundamental gap between spoken and signed languages in LLMs, we find that directly adapting previous paradigm to T2G is not ideal. Although reasoning LLMs benefit from strong generalization via pretraining on massive data, they inherently lack exposure to sign language glosses.
> This is evidenced by our attempt to apply the effective DeepSeek-R1-Zero for T2G, which only achieved 21.17 BLEU-4 on PHOENIX14T (as shown in Table 5).
> It is significantly lower than the baseline model.
> This result demonstrates that the unique characteristics of sign language gloss necessitate a customized solution rather than an off-the-shelf application. By explicitly leveraging the lexical overlap between languages, our approach effectively empowers a reasoning process for T2G.
> While the proposed approach might appear natural and straightforward, we submit that it constitutes a simple yet effective approach for T2G, addressing its challenges directly.
>
> **Q1:** Lack of Core Innovation: The method is essentially LLM + SFT on synthetic alignments + GRPO for refinement—a direct transplantation of GRPO to T2G. What is the essential technical departure from applying GRPO to any structured prediction task? Without a novel objective, architecture, or alignment mechanism, the contribution reduces to application, not advancement.
>
> **A1:**
> We agree that individual technical components (SFT and GRPO) are established.
> In our work, we introduce a different paradigm to improve T2G, which enhances translation with the designed reasoning mechanism.
> As you kindly pointed out in the strengths, our approach has been demonstrated effectiveness.
> The essential departure is rooted in the customized training objective and process tailored for T2G.
> Unlike general structured prediction tasks, T2G requires building a cross-lingual reasoning bridge from scratch within the pre-trained language model, as it possesses no prior sign language gloss knowledge.
> This challenge is evidenced by the failure of directly applying a powerful framework like DeepSeek-R1-Zero to T2G.
> This failure forced us to find a reasoning mechanism beneficial for T2G.
> Given the high cost of expert annotation, we devise an effective solution by leveraging the inherent lexical commonality between text and gloss to synthesize a customized reasoning process.
> However, we further identified that merely distilling knowledge from GPT-4o would create a performance ceiling. To break this bottleneck, we apply the GPRO algorithm to explore and improve upon the noisy initial alignments under the guidance of the final translation quality.
> Furthermore, its capability is complemented by pioneering performance on the challenging OOV problem (as the table below, Table 7 in the manuscript), aligned with our initial assumption.
> |Appearance|0(OOV)|$\leq 1$|$\leq 3$|$\leq 5$|$\leq 7$|$\leq 9$|
> |  --  |  --  |  --  |  --  |  --  |  --  |  --  |
> |Amount|14|10|39|60|72|86|
> |[Yao et al., 2024]|0|0|2.56|5.00|6.94|5.81|
> |Baseline|0|20.00|17.95|20.00|23.61|23.26|
> |T2G-Reasoner|14.29|30.00|28.21|26.67|29.17|26.74|

---

> ### Author Response · Authors · 2025-11-27
>
> **Q2:** Gloss-Dependent Generalization: The pipeline heavily relies on gloss-annotated data for both synthetic alignment generation and evaluation. On gloss-free benchmarks like OpenASL, the entire reasoning trace construction fails. How does the method adapt to real zero-shot or gloss-free SLT? This severely limits claimed generalizability.
>
> **A2:**
> Thank you for this insightful comment.
> We would like to clarify a potential misunderstanding.
> Our central contribution is to introduce a reasoning mechanism to improve translation quality in the current low-resource setting of T2G, as shown in our experiments on two public benchmarks, i.e., PHOENIX14T (7,096 samples) and CSL-Daily (6,598 samples).
> The generalizability we discussed and validated in our paper (i.e., ablation study on LLMs with different parameters) refers to the consistent performance improvement of our approach across base models of varying scales.
> We intended to show that our approach is a robust strategy and has the potential to achieve higher performance based on a better base model.
> Sorry for this confusion, we have revised it as 'scalability' in the manuscript.
>
>
> As for whether using gloss as an intermediate representation, we think that the cascade pipeline (i.e., text-to-gloss and gloss-to-gesture) is more achievable and practical for the application.
> Certainly, some more aggressive solution may emerge in the future like SORA2, which directly generates video through a diffusion model from spoken language descriptions. For now, given the low-resource scenario, a two-stage strategy is still a more achievable and practical solution in this field.
> We fully agree that exploring zero-shot or gloss-free sign language generation is a valuable research direction. In our future research, we would like to investigate pathways toward this goal, for instance, by exploring unsupervised alignment techniques or multi-modal learning.
>
> **Q3:** Sub-SOTA Performance  \& Weak Validation: On P14T-Dev, T2G-Reasoner (29.05) underperforms CV-SLT (29.27)—hardly a compelling result. No ablation across baselines (e.g., applying GRPO to CV-SLT, C2RL, etc.) leaves effectiveness unverified beyond the authors’ own setup.
>
> **A3:**
> Thank you for your insightful comments and pointing us to the relevant works [1,2].
> CV-SLT [1] and C2RL [2] achieve strong performance on SLT (generating spoken language texts from sign language video) on PHOENIX14T.
> They set a new standard for the video-to-text translation task, while we focus on the text-to-gloss translation (T2G) task.
> Since they are two fundamentally inverse and complementary tasks within a full translation system, we did not compare our approach with them.
> As shown in our Tables 1 and 2, our approach achieves improved translation performance over previous approaches on PHOENIX14T and CSL-Daily, as kindly pointed out by the Reviewer bJiD.
> As for the performance gap between these two different tasks, it also highlights a motivation for our work.
> The researchers pay more attention to the video-to-text translation task, while the text-to-video generation has received less attention and remains a significant challenge.
> Therefore, we focus on the first step (i.e., T2G) of the two-stage way (i.e., text-to-gloss and gloss-to-gesture) for sign language generation to support the development of SLG applications.
> As for applying our approach to CV-SLT [1] and C2RL [2], we agree that exploring the integration of our reasoning framework with powerful video-based baselines is a promising future direction. We are happy to investigate this application in the future, building upon the strong foundations [1,2].
> The above discussion has been added to the section of 'Conclusion and Future Work' in the revised paper.
>
> [1] Rui Zhao, Liang Zhang, Biao Fu, Cong Hu, Jinsong Su, and Yidong Chen. Conditional variational autoencoder for sign language translation with cross-modal alignment. In Proceedings of the aaai conference on artificial intelligence, 2024.
> [2] Zhigang Chen, Benjia Zhou, Yiqing Huang, Jun Wan, Yibo Hu, Hailin Shi, Yanyan Liang, Zhen Lei, and Du Zhang. C 2 rl: Content and context representation learning for gloss-free sign language translation and retrieval. IEEE Transactions on Circuits and Systems for Video Technology, 2025.

---

> ### Author Response · Authors · 2025-11-27
>
> **Q4:** Unexplained RL Collapse (Fig. 3C): Training with RL only (no SFT) yields drastically shorter thought chains. This suggests reward hacking or mode collapse, not efficient reasoning. Why does GRPO fail without SFT pre-alignment on SLT? Is KL control or reward shaping insufficient? This behavior undermines the robustness of the RL component.
>
> **A4:**
> We sincerely thank the reviewer for this insightful observation.
> The training curve for the RL-only setting (setting B in Table 5) shows a short reasoning length and poor translation performance.
> In most cases, the RL-only approach often generates English phrases unrelated to T2G (as examples shown in Appendix A.5). This directly leads to the observed short output length and translation failure.
> This is consistent with our hypothesis that without a foundational understanding of gloss tokens and syntax, the pre-trained language model lacks the necessary prior knowledge to explore meaningful reasoning paths via the same RL algorithm alone.
> It underscores that the inherent gap between spoken and signed languages makes it difficult for general-purpose LLMs to bootstrap reasoning from scratch.
> Our method provides a principled solution to this problem, and the stark contrast between Settings B and D in our results serves as direct empirical evidence for its necessity.
> We have added a discussion on this point in the revised manuscript (Sec. 4.3) to clarify the role of SFT as an essential enabler of RL-based reasoning.
> Thank you for your valuable discussion. We have added the above to the revision.

---

### Author Response · Authors · 2025-12-03
**Summary Comment of Contributions and Rebuttal Updates**

Dear Area Chair,

We sincerely thank you for your time and efforts in reviewing our paper.
We would like to provide a brief summary of our paper and the significant updates made during the rebuttal to assist your final decision.

*1. Core Contributions*

Our core contribution is to **introduce a reasoning mechanism to improve translation quality in the current low-resource scenario of text-to-gloss translation (T2G)**.
Sign language gloss is the transcription of sign language sign-by-sign, where each sign has a unique identifier.
The previous methods jointly align the embedding space of both languages in a data-driven manner.
Since the data collection and annotation of sign language requires specialized knowledge, obtaining a large-scale text-gloss dataset is time-consuming and expensive.
For example, the public datasets PHOENIX14T and CSL-Daily contain $7,096$ and $6,598$ samples, respectively.
Scarce training resources hinder T2G models from learning accurate cross-lingual alignments.
To this end, we are motived to bridge this granularity gap while mitigating data scarcity via a reasoning process. Different from general NLP tasks, incentivizing the reasoning capabilities for T2G is challenging due to the absence of gloss information in LLMs' pretraining. Based on the inherent lexical commonality between two languages, we provide the first reasoning framework for T2G. The optimization involves imitating the synthetic reasoning data (SFT) and encouraging the model to explore more accurate alignments (RL). Beyond achieving SOTA results, T2G-Reasoner is feasible to address **out-of-vocabulary (OOV) challenges**, which is overlooked in prior research.


*2. Key Rebuttal Updates \& New Experiments*

During the rebuttal period, we conducted extensive new experiments to address reviewers' suggestions. These additions have significantly strengthened the paper's rigor:

+ **Appendix A.5: Case Study in the RL-only setting**

  - Addresses: **Q4 of Reviewer my4X (Score: 6)** asked for an explanation on the RL-only collapse.

  - Result: This is consistent with our hypothesis that without gloss-specific expertise, the LLM struggles to explore meaningful reasoning paths via the same RL algorithm alone. We provided a detailed case study of the failure patterns and more analysis in the main text.


+ **Appendix A.6: Relationship between Word Alignment quality and Translation accuracy**

  - Addresses: **W2 of Reviewer bJiD (Score: 4)** requested this statistical analysis.

  - Result: Given the lack of word-level alignment annotations, we define the alignment quality as the proportion of glosses in the model's generated alignment that appear in the GT. The correlation coefficient $r$ is $0.54$, which indicates a positive correlation between them.


+ **Appendix A.7: Performance on Gloss-to-Text Translation**

  - Addresses: **W3 and Q2 of Reviewer bJiD (Score: 4)** required experiment results on G2T to explore "our approach's potential in sign language translation."

  - Result: We conducted our approach on G2T. Our approach achieves promising performance gains on G2T (baseline vs. ours: $25.40$ vs. $26.70$).


+ **Appendix A.8: Potential Risk of Generating Non-existing Glosses**

  - Addresses: **Q1 of Reviewer bJiD (Score: 4)** raised concerns about 'Non-existing glosses'.

  - Result: We calculated the rate of generated glosses that are non-existing. It is nearly identical between the baseline model (0.38\%, $13/3456$) and our T2G-Reasoner (0.37\%, $13/3478$).


+ **Appendix A.9: GPT-4o for T2G**

  - Addresses: **W1 of Reviewer ByJD (Score: 4)** suggested that T2G can be addressed with zero-shot/in-context examples.

  - Result: We conducted experiments based on the GPT-4o to evaluate both zero-shot and few-shot ($M$ similar retrieved examples using BM25) settings on CSL-Daily for T2G. Neither setting (Zero-shot: $5.12$ BLEU-4, 10-shot: $11.24$ BLEU-4) yielded a satisfactory quality. They indicate that LLMs lack a fundamental understanding of cross-lingual alignments for T2G. T2G requires building a cross-lingual reasoning bridge from scratch. It also highlights the core contribution of our paper. Our approach represents a promising advancement in addressing low-resource translation tasks.

  - *Note:* Reviewer ByJD initially stated in 'Questions' that "I'm open to having my opinion changed about the magnitude of contribution in the paper."

*3. Conclusion*

We are delighted that the reviewers find the paper to be well-written (Reviewer bJiD and ByJD), the method has the potential of generalizing/inspiring to other low-resource scenarios (Reviewer bJiD and ByJD), offering valuable insights into OOV behavior (Reviewer bJiD), and the experiments to be solid/convincing (All Reviewers).

We have provided experiments, analyses, and clarifications to respond to each reviewer's comments, respectively. We have now revised the paper accordingly.

We appreciate your consideration.

Best regards,

Paper23323 Authors

---

### Meta-Review · Area_Chair_4FZu · 2026-01-10

**Summary:**

The paper proposes T2G-Reasoner, which adapts GRPO (SFT + RL) for text-to-gloss translation by decomposing the task into word alignment and translation. Reviewers appreciated the clear presentation and empirical validation showing modest improvements. However, all noted that the core novelty is limited, contributions are mostly incremental, and generalizability is restricted by reliance on gloss-annotated datasets. While the experiments are sound, performance gains are small, and RL behavior lacks robustness, limiting broader impact and making the contribution insufficient for acceptance.

**Reviewer Concerns:**

The authors’ rebuttal addressed minor issues such as clarifying experimental settings, correlation between alignment and translation, and cultural/terminology corrections. Outstanding concerns remain regarding core technical novelty, reliance on gloss supervision, sub-SOTA performance relative to stronger baselines, limited broader significance beyond PHOENIX/CSL-Daily benchmarks, and unclear robustness of the RL component. Despite explanations, these key limitations persist, leaving reviewers unconvinced of the paper’s ICLR-level contribution.

**Reviewer Scores:**

If reviewers had been able to participate fully in the discussion, my assessment is that: Reviewer 1 would likely maintain a borderline “6” due to limited novelty; Reviewer 2 would stay near “4” as concerns about incremental contribution and limited generalizability remain; Reviewer 3 would also retain a “4” given the modest gains and dataset-specific impact. Overall, all reviewers’ opinions support a rejection, as the concerns on significance, novelty, and robustness are not fully resolved.

---

### Decision · Program_Chairs · 2026-01-26

Reject